# The Impact of Dietary Counselling on Achieving or Maintaining Normal Nutritional Status in Patients with Early and Locally Advanced Breast Cancer Undergoing Perioperative Chemotherapy

**DOI:** 10.3390/nu14122541

**Published:** 2022-06-18

**Authors:** Magdalena Jodkiewicz, Agnieszka Jagiełło-Gruszfeld, Agnieszka Surwiłło-Snarska, Beata Kotowicz, Małgorzata Fuksiewicz, Maria Małgorzata Kowalska

**Affiliations:** 1Department of Clinical Nutrition, Maria Skłodowska-Curie National Institute of Oncology-State Research, 00-001 Warszawa, Poland; agasurwillo@gmail.com; 2Department of Breast Cancer and Reconstructive Surgery, Maria Skłodowska-Curie National Institute of Oncology-State Research, 00-001 Warszawa, Poland; agnieszka.jagiellogruszfeld@gmail.com; 3Independent Laboratory of Cancer Biomarkers and Cytokines, Maria Skłodowska-Curie National Institute of Oncology-State Research, 00-001 Warszawa, Poland; beata.kotowicz@pib-nio.pl (B.K.); malgorzata.fuksiewicz@pib-nio.pl (M.F.); maria.kowalska@pib-nio.pl (M.M.K.)

**Keywords:** breast cancer, neoadjuvant chemotherapy, diet, body fat, obesity

## Abstract

**Background:** Obesity is an independent prognostic factor and is associated with poorer response to oncological treatment of breast cancer. Obesity is associated with shorter overall survival and shorter time to recurrence. **Material and methods:** The study included 104 breast cancer patients qualified for neoadjuvant chemotherapy. The control group consisted of 40 patients who refused to participate in the study. Consultation before chemotherapy included: author’s diet questionnaire, body composition analysis, nutrition education. After chemotherapy, the effects of the first dietary advice were evaluated. **Results:** More than half of all women had a BMI above normal before treatment. Analysis of the effects of nutrition education showed a significant improvement in body composition. After education, a slight increase in body weight and a significant decrease in fat mass and fat percentage were observed. In women who did not participate in education, a statistically significantly greater increase in body weight after chemotherapy was noted. Nutrition education of the study group did not prevent adverse changes in lipid profile resulting from chemotherapy. **Conclusions:** Dietary counselling prior to neoadjuvant chemotherapy may limit weight gain and may also influence fat mass reduction. Implementation of dietary recommendations does not guarantee maintenance of normal lipid parameters during chemotherapy.

## 1. Introduction

Breast cancer is the most common malignant tumour in women in Poland and worldwide. The highest incidence is reported between the ages of 50 and 74. The number increases significantly after the age of 35 [1,2].

According to a recent World Cancer Research Fund/American Institute for Cancer Research report, there is compelling evidence to suggest an increased risk of postmenopausal breast cancer in overweight and obese women in adulthood (after the age of 30). This risk increases with increasing body weight in adulthood [3]. It is known that adipose tissue is not only an energy store, but also performs the functions of thermal and mechanical protection, prevents the accumulation of fat in other tissues, and above all, displays multidirectional activity. The demonstration of the ability of adipocytes (the dominant cells of adipose tissue) to release active substances, so-called adipokines, acting not only locally but also on distant organs, made it possible to classify adipose tissue as an endocrine organ. The profile of adipokines released is not constant; it changes with the degree of obesity. Increased fat mass promotes the activity of factors with pro-inflammatory effects [4]. Studies on cancer formation are based on one of the oldest theories, i.e., Virchow’s theory, which points to chronic inflammation accompanying obesity as the cause [5,6,7]. 

Analysis of many clinical studies has indicated a relationship between obesity and an increased risk of breast cancer in both pre- and postmenopausal women [8,9,10]. Numerous studies have confirmed that women with second- and third-degree obesity had a higher risk of breast cancer than women of normal weight [11]. Additionally, a higher risk of the disease was observed in obese postmenopausal women [12]. The literature data also indicate an increased risk of breast cancer deaths in obesity [13,14]. Obesity is classified as an independent prognostic factor and is associated with a poorer response to oncological treatment. Furthermore, it has been observed that obesity is associated with shorter overall survival and shorter time to recurrence [15,16]. 

Studies in obese, postmenopausal women demonstrate increased risk of hormone-dependent breast cancer, which is explained by the indication of adipose tissue as the site of androgen aromatisation to oestradiol [17]. This source of oestrogens is confirmed by clinical studies indicating higher serum concentrations of oestradiol and estrone in obese postmenopausal women compared to their concentrations in lean women [14]. 

Literature data indicate that not only excess body fat but also its distribution is a significant problem. This was demonstrated in a study in women with a normal BMI who had a 56% increase in the risk of developing oestrogen-positive (ER+) breast cancer with a 5 kg increase in trunk fat mass, despite a normal BMI. Increased trunk fat was also associated with metabolic disorders and inflammation characterised by elevated blood levels of C-reactive protein and interleukin-6 [18]. Postmenopausal women with increased levels of body fat are also at increased risk of breast cancer, despite a normal BMI [18]. 

Primary prevention of breast cancer is limited. Modifiable factors include dietary factors such as prevention of overweight and obesity by avoiding the Western dietary model, reduction of alcohol consumption and elimination of active and passive smoking, and increased physical activity [19,20]. 

According to the World Cancer Research Fund/American Institute for Cancer Research, an unbalanced diet and poor eating habits are ten times more important risk factors for the development of many cancers than other aetiological factors [3]. 

The implementation of the present study is an attempt to answer the question of whether, at the first stage of breast cancer treatment, which is preoperative chemotherapy, dietary counselling can influence the achievement or maintenance of normal nutritional status. We know from the literature that weight gain, low physical activity and changes in lipid parameters during oncological treatment in this group of patients are a common problem; thus, the implementation of nutritional advice during neoadjuvant treatment seems necessary. An additional factor supporting the implementation of dietary counselling at this stage of treatment is the side effects of therapy, which can be reduced or even completely eliminated by an appropriately tailored diet [21,22]. 

## 2. Purpose and Scope of Work

The aim of this study was to determine the value of dietary advice in maintaining normal nutritional status or changing to normal nutritional status in patients with early and locally advanced breast cancer undergoing perioperative chemotherapy. 

## 3. Material and Methods

The study was conducted between 2017 and 2019. A total of 104 patients with early and locally advanced breast cancer undergoing preoperative chemotherapy, at the Maria Skłodowska Curie National Institute of Oncology in Warsaw, Poland, were enrolled in the study. The mean age of patients entering the study was 46 ± 13 years. The control group consisted of 40 patients who did not consent to participate in the planned study. These patients did not have a dietary consultation. Only body weight and calculated BMI before and after completion of neoadjuvant treatment were taken into account to compare nutritional status between the control and study groups. These data were routinely recorded on physical examination in the patient’s electronic medical history. The characteristics of the study and control groups, including diagnosis and type of treatment used, are described in Table 1. The study was approved by the Bioethics Committee of the Maria Skłodowska Curie Institute Oncology Centre in Warsaw No. 21/2017.

### 3.1. Inclusion Criteria

Inclusion criteria were female sex, age ≥ 18 years, histologically confirmed localized breast cancer and eligibility for neoadjuvant chemotherapy. Exclusion criteria were to contraindications due to the use of low-intensity electrical current during body composition testing and included patients with a cardiac defibrillator, metal implants, epilepsy and pregnant women.

The study included two dietary consultations: before starting and after completing chemotherapy. A schematic of the study (flow chart) is shown in Figure 1.

### 3.2. Scope of the First Dietetic Consultation:

Nutritional history;completion of the author’s questionnaire (Annex 2) on diet;evaluation of anthropometric parameters by means of body composition analysis with discussion of the results;analysis of laboratory lipid parameters;individual dietary advice on nutrition during chemotherapy.

### 3.3. Scope of the Second Dietetic Consultation:

Reassessment of anthropometric parameters by means of body composition analysis with discussion of the result;discuss nutritional problems (nausea, vomiting, diarrhoea, food aversion) with chemotherapy with possible Appendix A on nutrition after chemotherapy;re-examination of laboratory lipid parameters;refilling in the same original questionnaire on diet.

### 3.4. Nutrition Education

The nutritional education of the patients during the first dietetic consultation lasted up to 60 min and was based on recommendations for healthy eating with modifications due to possible gastrointestinal complaints aggravated during the treatment. The dietary recommendations were not a reduction diet. The energy and nutritional value of the diet was in accordance with the Standards of Nutrition in Oncology [21]. The main dietary recommendations given to the patients were:-regular consumption of 5 meals per day, including elimination of snacking between meals;-the recommended forms of heat treatment, including mainly the elimination of heat treatment with fat;-eliminating products that are sources of simple carbohydrates (sweets, confectionery, sweet dairy products, sweet drinks) and added sugar (sweetening drinks, sweetening salads, etc.)-eliminating fast food;-drink enough fluids (at least 2 litres/day);-eliminate alcohol consumption;-eating lean and unprocessed meat;-eating seafood (at least 2–3 times a week); -eating lean, natural dairy products;-eating vegetables (about 600 g) and fruit (200 g) except grapefruit and pomegranate.

### 3.5. Author’s Survey

The author’s questionnaire consisted of 12 closed-ended single-choice questions, including 10 questions about the amount of food consumed and the frequency of consumption of specific food groups, which were scored. The evaluation consisted of adding up all the points (maximum 20 points) from each question. No negative points were used in the evaluation of the survey. Each answer was scored according to the degree to which the dietary recommendations were implemented. 

### 3.6. Anthropometric Measurements

Anthropometric measurements in all patients were performed with a body composition analyser (Tanita Analizator BC 418 MA), using electrical bioimpedance.

### 3.7. Laboratory Parameters Lipid Metabolism

Venous blood was collected on an empty stomach, at least 12 h after the last meal. Determinations of total cholesterol, LDL, HDL fractions, and triglycerides were performed at the Department of Pathology and Laboratory Diagnostics, National Cancer Institute, Warsaw. 

### 3.8. Statistical Analysis

In the statistical analyses, the Shapiro–Wilk test was used to check the consistency of the distribution of each variable with the normal distribution; Student’s t test and the Wilcoxon test were used to assess the statistical significance of the changes. Spearman’s non-parametric correlation coefficient was used to assess the relationship between data. A significance level of *p* < 0.05 was assumed for all tests. Statistical analysis was performed using RStudio software (RStudio Team (2020). RStudio: Integrated Development for R. RStudio, PBC, Boston, MA, USA URL http://www.rstudio.com/, accessed on 1 July 2020). 

## 4. Results—Description of the Study Group

### Analysis of Anthropometric and Laboratory Measurements before Treatment

The first dietary consultation before neoadjuvant treatment was attended by 104 patients, including 72 patients (69%) premenopausal and 32 patients (31%) postmenopausal women. Due to the different hormonal status that may influence the results of the measurements, e.g., increased body fat production, the results are also presented taking into account hormonal status.

Among all subjects, more than 55% had an abnormal BMI, of which a higher proportion were overweight patients. Among premenopausal women, more than 45% had excessive body weight, and in postmenopausal women, the percentage was more than 78%. Analysis of body composition in the entire pre-treatment group showed that more than 40% of the women entering the study had an elevated body fat percentage, including more than half of the postmenopausal women who had an elevated body fat percentage before treatment (Table 2).

The parameters of percentage body fat, fat mass, lean body mass and total body water content were statistically analysed taking into account the history of menopause. Postmenopausal women had significantly higher body fat mass and body fat percentage. The mean body water content in the whole group was significantly below the recommended value (Table 3).

Of the lipid metabolism parameters determined, total cholesterol concentrations were statistically higher in postmenopausal women. Other parameters remained within the reference limits (Table 4).

## 5. Results—Changes after Treatment

### 5.1. Analysis of Parameters Related to Nutritional Status (Body Composition Analysis and Lipid Metabolism Laboratory Results) before and after Chemotherapy

After completion of chemotherapy, 75 patients attended the second dietary consultation, which was 72% of the patients who attended the first dietary consultation. Among the participants, 29% were postmenopausal women and 71% were premenopausal. 

The percentage of body fat and body fat mass were significantly reduced, and there was a statistical increase in lean body mass and total water content (Table 5).

Statistical analysis of lipid metabolism parameters showed a significant increase in mean total cholesterol, LDL cholesterol and triglycerides after completion of chemotherapy compared to baseline values. The mean HDL cholesterol concentration decreased significantly. (Table 6).

### 5.2. Results of a Questionnaire on Diet before and after Chemotherapy

The maximum number of points that could be obtained in the questionnaire was 20 pts, an increase in the number of points scored in the questionnaire after nutrition education and treatment meant a favourable change in the patients’ diet (Table 7). 

Statistical analyses of correlations of parameters determined after completion of neoadjuvant treatment did not confirm correlations between the change in diet and parameters of body composition analysis and percentage body fat and concentrations of lipid metabolism parameters.

### 5.3. Comparison of Changes in BMI and Body Weight Values between Control and Study Groups

There were significant differences in weight gain after chemotherapy between the control group without nutritional education and the study group (Table 8).

## 6. Discussion

The problem of overweight and obesity among breast cancer patients is well documented and is not related to hormonal status, but is more common in postmenopausal women [15,23]. The present study shows that more than half of all women who participated in the study had a BMI above normal. The problem of obesity, as a risk factor, is treated differently in postmenopausal women and has a different significance in premenopausal women. The literature data confirm that in postmenopausal women the risk of estrogen-positive breast cancer increases with every increase in BMI by 5 kg/m2. In premenopausal women, a negative correlation between obesity and the risk of oestrogen-positive cancer was most commonly found, while obesity was associated with an increased risk of oestrogen-negative and triple-negative breast cancer. However, regardless of hormonal status, obesity is a poor prognostic factor for overall survival time, and dietary change and increased physical activity may contribute to prolonged survival [24,25,26,27]. In our study, adequate to the elevated BMI, an elevated adipose tissue content was observed, which concerned mainly postmenopausal patients. A similarly high percentage of adipose tissue before the start of treatment was observed by other authors [28,29,30]. In our study, lipid metabolism parameters were measured, which showed statistically higher mean total cholesterol concentration in the group of postmenopausal women, which was 207.7 mg/dL. Mean LDL cholesterol and triglyceride concentrations remained below the cut-off points in both patient groups. Similar baseline values of lipid parameters were reported by other authors of a study in which patients before neoadjuvant chemotherapy had total cholesterol levels at the borderline of normal at 199 mg/dL, while other lipid parameters were normal [31]. However, lipid metabolism parameters do not always remain within normal limits [32]. 

Clinical observational reports on changes in body composition and lipid metabolism-related parameters under neoadjuvant chemotherapy inspired us to ask whether providing dietary care to patients early in treatment could help maintain or achieve normal nutritional status and optimal lipid parameters. Unfortunately, most scientific studies did not include dietary intervention, and according to the literature, an increase in body weight after chemotherapy was observed in these patients, which was also observed in our study in an uneducated control group [28,33,34,35]. There are also scientific reports that deny weight changes during chemotherapy, even without nutritional intervention [36]. Despite the discrepancy in findings, authors agree that excessive body weight in women diagnosed with breast cancer is an adverse prognostic factor [37].

Our study analysed the change in parameters obtained from body composition analysis and changes in lipid metabolism after chemotherapy in subjects receiving nutritional intervention. In the results, a slight increase in body weight was observed in the study group, and it was significantly lower compared to other studies in which nutrition education was not provided. In our own study, changes were also observed in body composition, in which there was a significant decrease in fat mass and fat percentage. Similar results were obtained by the authors of another study in which an individualised nutritional intervention programme was implemented during treatment [38]. Adverse changes in body composition were reported for studies in which patients did not receive dietary care. Authors have reported a significant increase in fat mass in both pre- and postmenopausal women [28,34,36,39]. An interesting phenomenon is the increase in percentage of body fat with no change in body weight and BMI that has been reported in premenopausal women [39]. In studies by other authors, both with and without implemented nutritional intervention, a decrease in lean body mass was observed [38,39]. While the authors of these papers suggest that as a result of chemotherapy, changes in sex hormone production may mimic the physiological changes associated with menopause that cause fat accumulation and a decrease in lean body mass, a beneficial increase in lean body mass by an average of 1.2 kg was noted in our study. Beneficial changes were also noted in total body water content, which increased by an average of 1.2% after chemotherapy. In the studies discussed above in which electrostatic bioimpedance was performed, only Jung et al. used this parameter in describing the results, indicating that there were no significant changes [36].

In our study, similarly to other authors, despite the applied nutritional intervention, adverse changes in lipid parameters were observed as a result of chemotherapy. In this study, we observed a significant increase in total cholesterol, LDL cholesterol and triglycerides and a decrease in HDL cholesterol. The results obtained in our group are consistent with the findings of other authors [31,40,41]. The mechanism of these changes is not fully known. The probable reason is the association of lipid metabolism with sex hormones. Changes in lipid profile correlate with menstrual changes, which are an effect of chemotherapy [42,43]. Another mechanism discussed is treatment-induced oxidative stress, which causes changes in lipid metabolism [44].

The main aim of the study was to assess the impact of nutrition education on achieving or maintaining normal nutritional status. A proprietary dietary assessment questionnaire, based primarily on the frequency of intake of specific food groups, was used to assess the effectiveness of nutrition education. Questionnaire scores obtained after nutrition education and completion of chemotherapy indicated a significant improvement in diet compared to the same questionnaire completed before nutrition education and treatment. The mean total scores before and after nutrition education were statistically significantly different. The changes in the questionnaire mainly concerned the complete elimination of alcohol from the diet, which was applied by the vast majority of patients. In addition, after nutritional education, the patients increased the amount of vegetables consumed, eliminated or reduced the amount of sweets consumed and reduced the consumption of fried foods. There was also a big change in meat consumption. The patients abandoned pork in favour of poultry and, less frequently, beef. The results obtained in our study are similar to those obtained by Anderson et al., who on the basis of the results obtained from the food frequency questionnaire and the calculated index of overall diet quality, noted that the women studied increased their fibre intake and decreased the percentage of energy from fat in their diet. In addition, an improvement in the diet quality index was noted in this group, with a reduction of more than 5% of body weight [45].

Based on the difference in scores obtained in the author’s questionnaire, a correlation analysis was conducted between the change in diet and parameters related to nutritional status obtained from body composition analysis, where no statistically significant relationships were demonstrated. A similar analysis was carried out in relation to lipid parameters, where also no statistically significant differences were found. No correlation was found between the percentage change in body fat and lipid parameters. However, a trend towards a decrease in total and LDL cholesterol was observed with a statistically significant change in percentage body fat before and after chemotherapy.

Research in the area of dietetics is often problematic. One point of contention is the selection of an appropriate tool to assess dietary intake. The main limitation of a reliable assessment of the subjects’ diets is the possible falsification of answers. Another problem of research in the area of human nutrition is the selection of a control group. In the case of studies involving people with a specific disease entity, it seems unethical not to provide nutritional education to patients who have agreed to take part in the study. Consequently, a large proportion of studies do not mention a control group. Analysing the world literature, it has been noticed that few authors decide to compare the studied group of people with malignant tumours to a healthy population [28]. The use of the healthy population as a control group for subjects with a cancer diagnosis seems to be inadequate. Therefore, in our study, it was considered that the control group could be women with the same diagnosis and treatment who did not agree to participate in the study. In order to compare these groups, it was only possible to compare changes in body weight and BMI, which are routinely completed in medical records. Analysis of these data showed that the increase in body weight in the control group was statistically greater compared to the group that had received nutritional education before treatment.

The results of the submitted study indicate the important role of dietary counselling in the treatment of breast cancer already at the first stage, which is neoadjuvant chemotherapy. Providing information on the impact of diet on treatment effects concerns such important elements such as: maintaining or reducing body weight, obtaining favourable changes in body composition, even in women with normal BMI, and reducing gastrointestinal complaints, which are side effects of oncological treatment.

Dietary consultation is a relatively low cost to the hospital and can significantly improve patient outcomes and quality of life.

## 7. Conclusions

Nutrition education makes a significant difference in changing diets.

Dietary counselling prior to systemic treatment may limit weight gain during chemotherapy and can have an effect on reducing fat mass.

Implementation of dietary recommendations does not guarantee maintenance of normal lipid parameters during chemotherapy.

## Figures and Tables

**Figure 1 nutrients-14-02541-f001:**
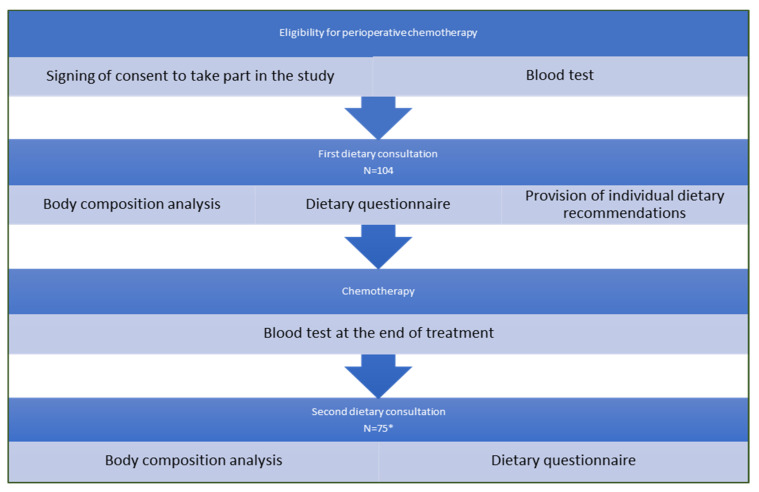
Flow chart—schematic of the study. * All participants in the study survived and completed chemotherapy, and the proportion of patients who did not return after treatment was due to reasons beyond the study design. We suspect that in some participants, the reasons were ill health and distance of residence from the treatment centre.

**Table 1 nutrients-14-02541-t001:** Characteristics of the study and control group.

Variable	Parameter	Dietary Consultation Group		Control Group	
		N = 104	%	N = 40	%
**cT**	0	1	1	0	0
	1	8	7.7	4	10
	2	58	55.7	23	57.5
	3	34	32.7	12	30
	4d	3	2.9	1	2.5
**cN**	0	49	47.2	10	25
	1	46	44.2	25	62.5
	2	5	4.8	4	10
	3	4	3.8	1	2.5
**G**	1	11	10.5	3	7.5
	2	65	62.5	29	72.5
	3	28	27	8	20
**ER**	positive	40	38.5	12	30
	negative	64	61.5	28	70
**PR**	positive	32	30.8	11	27.5
	negative	72	69.2	29	72.5
**HER2**	positive	15	14,4	4	10
	negative	89	85.6	36	90
**Type of neoadjuvant** **treatment**	TCarbH	15	14.4	4	10
	ACdd/ PCL weekly	45	43.3	19	47.5
	AC/PCL weekly	44	42.3	17	42.5
**Characteristics of achieved responses to treatment**	pCR	64	61.5	22	55
	Non-pCR	40	38.5	18	45

cT—clinical tumour; cN—clinical nodes; G—grade; ER—oestrogen receptor; PR—progesterone receptor; HER2—human epidermal growth factor receptor 2; TCaH—docetaxel, carboplatin, trastuzumab; ACdd/PCL—doxorubicin, cyclophosphamide dose dense (every 2 weeks), paclitaxel; AC/PCL—doxorubicin, cyclophosphamide (every 3 weeks), paclitaxel; pCR—pathological complete response; non-pCR—non pathological complete response.

**Table 2 nutrients-14-02541-t002:** Distribution of BMI and percentage body fat in the study group of patients.

	All N = 104	Before Menopause N = 72	After the Menopause N = 32
BMI: <18.49	3 (2.9%)	3 (4.2%)	0 (0%)
BMI: 18.5–24.99	43 (41.4%)	36 (50%)	7 (21.9%)
BMI: 25.0–29.99	33 (31.7%)	17 (23.6)	16 (50%)
BMI: ≥30	25 (24%)	16 (22.2%)	9 (28.1%)
%Fat—normal	54 (51.9%)	41 (57%)	13 (40.6%)
%Fat above normal	43 (41.4%)	25 (34.7%)	18 (56.3%)
% Fat below normal	7 (6.7%)	6 (8.3%)	1 (3.1%)

**Table 3 nutrients-14-02541-t003:** Mean concentrations of body composition analysis parameters according to hormonal status.

	All N = 104		Before Menopause N = 72		After the Menopause N = 32		*p* Value
	Average	SD	Average	SD	Average	SD	
**Fat (%)**	31.6	7.4	29.8	7.1	35.2	7.0	**0.0054 ***
Fat mass (kg)	22.2	8.3	20.6	7.9	25.3	8.3	**0.0279 ***
FFM—Fat-free Mass (kg)	45.4	5.9	45.6	6.4	45.0	5.0	0.540
TBW—Total body water (%)	33.5	3.6	33.8	3.6	32.9	3.7	0.458

SD—standard deviation *—*p* < 0.05.

**Table 4 nutrients-14-02541-t004:** Mean levels of lipid metabolism parameters before treatment by hormonal status.

	All N = 104		Before Menopause N = 72		After the Menopause N = 32		*p* Value
	Average	SD	Average	SD	Average	SD	
**Total cholesterol mg/gL**	188.2+	46.5	178.5	45.9	207.7	41.9	**0.0219 ***
HDL (mg/dL)	53.9	14.4	52.3	15.8	57.0	10.9	0.05
LDL (mg/dL)	113.1	33.7	110.2	31.0	119.5	39.2	0.54
TG (mg/dL)	107.7	60.8	99.7	52.9	124.1	72.9	0.138

SD—standard deviation *—*p* < 0.05.

**Table 5 nutrients-14-02541-t005:** Changes in the results of anthropometric parameters of body composition analysis before and after chemotherapy.

N = 75	Before		After		Change		*p* Value
	Average	SD	Average	SD	Average	SD	
**Body weight (kg)**	68.2	12.0	68.4	11.5	0.1	4.4	0.671
BMI (kg/m^2^)	25.3	4.4	25.5	4.3	0.1	1.4	0.274
Fat (%)	31.6	7.4	30.2	6.8	−1.8	3.9	**<0.001 ***
Fat mass (kg)	22.2	8.3	21.4	7.8	−1.2	3.5	**0.018 ***
FFM—fat-free mass (kg)	45.4	5.9	47.1	5.9	1.5	5.4	**<0.001 ***
TBW—total body water (%)	33.5	3.6	34.8	3.5	1.2	1.4	**<0.001 ***

SD—standard deviation *—*p* < 0.05.

**Table 6 nutrients-14-02541-t006:** Changes in concentration of laboratory parameters before and after chemotherapy.

N = 75	Before		After		Change		*p* Value
	Average	SD	Average	SD	Average	SD	
**Total cholesterol (mg/dL)**	188.2	46.4	196.6	44.4	8.9	48.4	**0.019 ***
HDL (mg/dL)	53.9	14.4	50.7	11.0	−3.3	10.4	**0.027 ***
LDL (mg/dL)	113.1	33.7	124.4	54.3	12.6	52.5	**0.019 ***
TG (mg/dL)	107.7	60.8	146.8	84.9	42.1	74.6	**<0.001 ***

SD—standard deviation *—*p* < 0.05.

**Table 7 nutrients-14-02541-t007:** Dietary assessment based on scores from the author’s questionnaire before education and treatment and after treatment.

N = 75	Before		After		Change		*p* Value
	Average	SD	Average	SD	Average	SD	
**Survey points**	7.4	3.5	12.2	3.3	4.8	4.4	**<0.001 ***

SD—standard deviation *—*p* < 0.05.

**Table 8 nutrients-14-02541-t008:** Changes in body weight and BMI values before and after chemotherapy in the control and study groups.

	Control Group N = 40		Study Group N = 75		*p* Value
	Average	SD	Average	SD	
**Body weight delta (kg)**	1.8	3.8	0.1	4.4	* **p** * **= 0.023 ***
Delta BMI (kg/m^2^)	0.7	1.4	0.1	1.4	0.05

SD—standard deviation *—*p* < 0.05.

## Data Availability

Not applicable.

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
