# Peer review of "The Impact of Dietary Counselling on Achieving or Maintaining Normal Nutritional Status in Patients with Early and Locally Advanced Breast Cancer Undergoing Perioperative Chemotherapy"

_nutrients, 2022, doi:10.3390/nu14122541_

Round 1
Reviewer 1 Report
1-An interesting and underappreciated link between obesity and BC.
2-Why was it important to study this link in neoadjuvant therapy patients of BC?
3-After education, a slight increase in body weight and a significant decrease in weight and fat percentage were observed. How are these outcomes not mutually exclusive?
4-The results outlined in the abstract are confusing to some extent since it is not clear what study groups gave what outcome.
5- Sentences 39-40, these sentences are slightly confusing, is there a link between postmenopausal BC patients who were obese during adulthood or are currently obese? Also, is the definition of adulthood accurate (after 30) or is this the figure obtained from the study?
6- The profile of adipokines that correlates with the age of the women in this study would be interesting to include in your introduction.
7- Why is there a higher risk of BC in post compared to pre-menopause?
8- The logical flow of the content in paragraphs 49-57 is not great, some sentences would do better appearing first. For example, in lines 54-56 the classification of obesity could be moved up before the elaborate details about menopause and BC risk to follow. This will go from a wider definition to a more focused menopausal view if that makes sense.
9- The link between Virchow theory and inflammation could be moved up or consolidated with the adipokine section.
10- Also lines 73-79 seem like introductory sentences better suited to earlier sections of the introduction, they are currently breaking the flow. Once general information is given, then the authors could narrow in on the topic of interest. The logical flow of the topics can be better arranged (to avoid backward and forwards moving between topics). But it is up to you.
11- It is not clear how the 40 patients who did not give consent to take part in the planned study were recruited anyways? Please kindly recheck your ethical consent forms and amend this section.
12- Please clarify what tests were conducted in line 115. Were these linked to the tests stated in 141-144?
13- Line 120, what nutritional problems have occurred, please provide some examples. How much time elapsed between the first and second consultation?
14- In line2 125-136 please provide your questionnaire or any methods/ material used in a supplementary file to enhance reproducibility.
15- The choice to split the results into pre-and post-menopause in the tables is useful, and so is using bold fonts for any significant result.
16- In lines 188-189, what is the significance of changes to anthropometric parameters before and after treatment? Does this also tally with pre and post-manopause? Also, this question applies to lines 193-196 and the linkage between lipid metabolism and these parameters with pre and post-manopause.
17- Was there a link between education on diet, chemotherapy, and pre and post-manopause? Also weight gain in lines 210- 214?
18- A solid link (evidenced by statistical tests would be useful) to associate menopausal status with changes to anthropometric parameters/ lipid metabolism, education, and weight gain.
The discussion is comprehensive, thanks.
Author Response
1- The lack of dissemination of information on the relationship between modifiable obesity and BC is surprising.
2- Weight gain during oncological treatment in this group of patients is a very frequently reported problem both by doctors and the patients themselves. Often dietary consultations take place only after a large increase in body weight, even approx. 20kg. Unwanted weight gain is most often explained by reduced physical activity. Breast cancer patients undergoing treatment do not know how important it is to maintain or achieve a proper nutritional status during and after oncological treatment. They are also unaware of their exposure to weight gain due to the treatment applied. It was also important to learn about the type of atypical diets implemented particularly frequently in this group of patients. Neoadjuvant treatment may cause a number of discomforts, including gastrointestinal problems. To date, there are no standards of routine dietary care for patients diagnosed with breast cancer. The present study is an attempt to answer the question whether dietary consultation at the first stage of breast cancer treatment, i.e. preoperative chemotherapy, can affect the achievement or maintenance of normal nutritional status.
3- There is an error in the sentence. Correctly the sentence should read: After education, a slight increase in body weight and a significant decrease in fat mass and fat percentage were observed.
Changes have been inserted in the text.
4- In the summary in the results section, the first sentence refers to the study group in general, before the nutrition education. The second and third sentences refer to the study group after education, and the fourth sentence refers to the control group, i.e. without nutrition education. The last sentence also refers to the group after nutrition education, but it summarises the issues of lipid profiles, not body composition, which is why it is at the end of the results.
5- Citing a World Cancer Research Fund source / American Institute for Cancer Research report such an age was defined as adulthood. This is not the figure obtained in the study.
To make the sentence clearer it can be divided into two parts:
According to a recent World Cancer Research Fund / American Institute for Cancer Research report, there is compelling evidence to suggest an increased risk of postmenopausal breast cancer in overweight and obese women in adulthood (after the age of 30). This risk increases with increasing body weight in adulthood [3].
Changes have been inserted in the text.
6- Adipocytes are the predominant cells in adipose tissue. The demonstration of the ability of adipocytes to release active substances, so-called adipokines, acting not only locally, but also on distant organs, made it possible to classify adipose tissue as an endocrine organ. In women, the activity and distribution of adipose tissue is linked to hormonal status, i.e. it changes with age. Before menopause, the locus of adipose tissue is the buttock-femoral area, corresponding to the presence of the oestrogen receptor alpha. Decreasing estrogen concentrations, increasing androgens, contribute to the displacement of adipose tissue and an increase in visceral obesity. Visceral fat is more metabolically active than subcutaneous fat. There is an imbalance in the production of adipokines in favour of an increase in the production of pro-inflammatory factors. We observe an increase in leptin concentration and a decrease in adiponectin. The degree of change is influenced by obesity.
7- Studies of obese, postmenopausal women show an increased risk of hormone-dependent breast cancer compared to premenopausal women. One of the mechanisms explaining this relationship is the identification of adipose tissue as the site where aromatisation of androgens to oestradiol takes place. This source of estrogen is confirmed by clinical studies showing higher serum concentrations of estradiol and estrone in obese postmenopausal women than in lean women.
8- In lines 54-56 no classification of obesity is given. This sentence highlights that the risk of BC increases with the degree of obesity. This sentence is an expansion of the earlier lines.
In lines 49-54, which belong to the earlier paragraph, the subject of adipocytes and adipkoins is discussed.
9- Thank you very much for your comment. As suggested, the link between Virchow's theory and inflammation has been moved up and consolidated with the section on adipokines.
Changes have been inserted in the text.
10- This section deals with changes in body weight, BMI and body fat distribution. It is a kind of summary and a combination of the themes of the parts of the paragraphs above.
11- Patients were qualified for the study during routine medical examinations. Forty patients did not agree to participate in the study. They constituted the control group. These patients did not have a dietary consultation, did not have a body composition analysis performed and did not fill out the documentation that was part of the study. Therefore, only body weight and calculated BMI before and after neoadjuvant treatment were taken into account to compare nutritional status between the control and study groups. These data were routinely recorded on physical examination in the patient's electronic medical history.
Changes have been inserted in the text.
12- In line 115 the studies listed are re the same as in lines 141-144.
Changes have been inserted in the text.
13- The main nutritional problems discussed during chemotherapy were nausea, vomiting, diarrhoea and food aversion (changes inserted in text). The time between the first and second consultation depended on the duration of chemotherapy and was approximately 4-6 months. The discrepancy was due to different chemotherapy regimens and the possibility of postponement of individual chemotherapy cycles.
14- In the file we send the author's questionnaire used for our study. The questionnaire also contains questions that are not discussed in this publication. These relate to the use of alternative methods.
15- The division of results and bolding of significant results was intended to make the results easier to understand.
16- Changes in anthropometric parameters before and after chemotherapy did not distinguish between pre- and post-menopausal periods, as all patients entered menopause during treatment. The study did not track differences taking into account the division used at the beginning to characterise the study group. The reason for this was also the small group of postmenopausal women. In our study we were only interested in the overall results after education and treatment.
In our study, in the group of women studied, the differences in lipid profile between pre- and post-menopausal women concerned only total cholesterol. There were no differences in HDL, LDL and TG concentrations. In this case, as well as in the anthropometric measurements, all patients were postmenopausal after the study and we did not consider the distinction due to hormonal status for the post-treatment results.
17-At the start of treatment, all patients were entering menopause. Postmenopausal women represented a small proportion. Differences were not tracked in the study considering the hormonal status division used at the beginning. It was only used to characterise the study group.
18-After starting chemotherapy, all patients were put into menopausal status. We used menopausal status only to characterise the general group. We did not take education into account in the present study. The educational status of both the study group and more so the control group is unknown.

Reviewer 2 Report
Reviewer’s comments on an article of nutrients-1750604-peer-review-v1
Major comments
This is an article on study of an effects of nutritional education on anthropometry and lipid chemical profiles of patients with early and locally advanced breast cancer after adjuvant chemotherapy. The authors concluded that an nutritional education had a positive impact on body composition, body weight, an maintaining fat percentage. However, I must indicate several problems as the follows:
1. No explanations of nutritional education: what education they did, such as consultation or education of changing eating pattern, and how many minutes they consumed for each subject.
2. No data of characteristics before and after chemotherapy of two groups: control and study group.
3. No information of objective daily intakes of energy, protein, % energy of fat, and saturated fat that might have impact on lipid profiles.
4. No information of questionnaires that the authors used to evaluate foods the subjects had eaten. And were questionnaires validated or not? If no, the questionnaires must be validated first and then the questionnaires could be used for the study.
5. Without comparisons of chemotherapy regimen and interval how many days the patients had been taken as the adjuvant chemotherapy, it is unclear whether the anthropometric improvement might be results of nutritional intervention or chemotherapy itself.
As the study methods were not subscribed in detail such as the definition of nutritional education and its contents as aforementioned and no information of clinical severities of breast cancers (comparisons of clinical stages) and severities of chemotherapeutic regimens the patients had been taken, no association between the nutritional education and anthropometric parametric maintenance could be concluded.
In conclusion, I must say that this article could not be accepted without major revision.
Minor comments
1. In table 4, FFM free fat mass ⇒fat free mass
2. In line 105, any explanations of objective examples of contraindication to use the low-intensity electric current?
3. In 126, what is objective definitions of “healthy diet” with scientific evidences?
4. How about sex hormonal profile to explain the changes of lipid profiles? Did they measure serum concentrations of sex hormones before and after chemotherapy? If no, the changes of lipid profiles might be changes of sex hormone after chemotherapies.
Author Response
Major comments
1. The nutritional education of the patients during the first dietetic consultation lasted up to 60 minutes and was based on recommendations for healthy eating with modifications due to possible gastrointestinal complaints aggravated during the treatment. The dietary recommendations were not a reduction diet. The energy and nutritional value of the diet was in accordance with the Standards of Nutrition in Oncology. The main dietary recommendations given to the patients were:
-regular consumption of 5 meals per day, including elimination of snacking between meals;
-the recommended forms of heat treatment, including mainly the elimination of heat treatment with fat;
-eliminating products that are sources of simple carbohydrates (sweets, confectionery, sweet dairy products, sweet drinks) and added sugar (sweetening drinks, sweetening salads, etc.)
-eliminating fast food;
-drink enough fluids (at least 2 l/day)
-eliminate alcohol consumption
-eating lean and unprocessed meat
-eating seafood (at least 2-3 times a week)
-eating lean, natural dairy products
-eating vegetables (about 600g) and fruit (200g) except grapefruit and pomegranate
and pomegranates.
Changes have been inserted in the text.
2. Data on the characteristics of the study and control groups are described in the table and included in the text.
3. The intake of protein, fat and carbohydrates was recommended at the level recommended in the Polish Standards of Nutrition in Oncology, i.e. protein 15-20%, fat 30-35%, carbohydrates 45-55%. In this study, a food frequency questionnaire was used to assess changes in nutritional status, which makes it impossible to calculate the exact percentage of fat in the patients' diets. Despite a significant improvement in diet assessed by the author's questionnaire, including a reduction in the consumption of fatty meat and its products, the lipid profile changed unfavourably, which is also confirmed by the authors of other studies.
4. The questionnaires used for the study were not validated, due to the impossibility of repeating the survey under the same conditions. The first questionnaire was conducted before the start of chemotherapy. Then the treatment began, during which time complications arose and changes occurred due to nutritional education as well as the treatment.
5. The treatment regimen depended on the biological subtype of the cancer. Patients with triple-negative cancer and luminal carcinoma A or B received 4 courses of AC (doxorubicin + cyclophosphamide) every 3 weeks followed by 12 courses of PCL (paclitaxel) every week. In this case, two regimens were possible ( with AC given every2 weeks (dose dense) or every 3 weeks. On the day of chemotherapy administration, steroids were administered (dexaven 8 mg iv - as nausea/vomiting prophylaxis). In patients with confirmed overexpression of the human epidermal growth factor receptor 2 (HER2) 6 courses of TCH (docetaxel + carboplatin + trastuzumab) were administered. In this group of patients, steroids were administered the day before chemotherapy administration, the day of and the day after chemotherapy administration (8 mg p.o. each). HER+ patients made up a small proportion (14.5% in the study group).
As a dietitian, I only knew the diagnosis and focused on nutrition education. I knew what the diagnosis was and that all patients were qualified for neoadjuvant chemotherapy.
The control group was subjected to the same treatment conditions as the study group, so the treatment time was also comparable. The only difference between the study group and the control group was whether or not they received nutritional education.
Minor comments
1.Changes have been inserted in the text.
2. According to information from the manufacturer of the Tanita body composition analyser, the test is not recommended for patients with an implanted cardiac defibrillator, metal implants, as well as patients with epilepsy and pregnant women. Such information is included in this paragraph.
3. Dietary orders given to patients were in accordance with the Polish Standards of Nutrition in Oncology, taking into account individual gastrointestinal complaints. The principles of a healthy diet were in accordance with the Pyramid of Healthy Eating and the current model of the Healthy Eating Plate.
4. The sex hormone profile was not investigated in this study. The mechanism of lipid profile changes may be dependent on the sex hormone profile. The only conclusions drawn in the study were that it was not possible to maintain a normal lipid profile as before chemotherapy or to limit its changes with a healthy diet.

Round 2
Reviewer 1 Report
The authors have addressed my comments.
Author Response
Thank You for your review.
Reviewer 2 Report
Review comments on an article of nutrients-1750604-peer-review-v2
Major comments
This is a revised article on effects of nutritional education on patients before chemotherapy of patients of breast cancer. The authors concluded that a nutritional education before chemotherapy could have impacts of maintaining body weight and decreases of body fat after chemotherapy comparing with control group in that patients did not receive nutritional education. These conclusions seems to be related with an effect of nutritional education, because the authors compared these effects between intervention and control group with and without nutrition education. However, it seems unclear how many subjects were involved in each group observed in table 2 – 8. In addition, the authors must add the study flow chart to explain how many subjects could complete chemotherapy and survived even after 2nd dietetic consultation in the intervention group because it is unclear whether all subjects could survive even with advanced breast cancer after chemotherapy and reached to second dietetic consultation.
In conclusion of my reviewing results, the article must be added with flow chart figure of study methods and the numbers of each group shown in table 2-8 before final version.
Minor comments
1. The authors must divide their results into at least two parts: part 1, comparison between pre- and post-menopause at the parts of table 2 3, and 4: part 2, comparison between before and after chemotherapy at the parts of table 5, 6, and 7. This article seems not so easy to understand because of complex structures of the article.
2. In table 8, body mass in third line must be corrected to body weight because body mass was named body weight in the title of this table.
3. In also table 8, test group must be corrected to study group because it appeared in title of this table also.
4. The authors must explain the reason why they identified healthy eating in line 159 for listed them from line 171 to line 178. Otherwise the readers might be unaware the reasons why the authors used the words of healthy eating and the words of “healthy eating” seems a priori and not scientific.
Author Response
Major comments
In the text and in Tables 2-8 it was added how many patients there were in total, and in Tables 2-3 by how many before and after menopause.
The Results- changes after treatment section describes how many participants returned for a second dietary consultation after chemotherapy. All participants in the study survived chemotherapy, and the proportion of patients who did not return after treatment was due to reasons outside of the study design. We suspect that in some participants, the reasons were ill health, and distance of residence from the treatment centre.
A flow chart of the study and the numbers of each group in the tables have been added to the article.
Minor comments
1-The results were divided into two parts: "Results- description of the study group" and "Results- changes after treatment"
2-Changes have been inserted in the text.
3- Changes have been inserted in the text.
4- The term healthy eating has been used to represent in general terms the recommendations given to patients. These recommendations were based on the model of the Healthy Eating Plate developed by the National Centre for Nutrition Education in Poland and issued under the auspices of the Polish Ministry of Health. In addition, the dietary recommendations recommended to patients were in accordance with the Polish Standards of Nutrition in Oncology, taking into account individual gastrointestinal complaints.
